# Canadian Free Sugar Intake and Modelling of a Reformulation Scenario

**DOI:** 10.3390/foods12091771

**Published:** 2023-04-25

**Authors:** Jodi T. Bernstein, Anthea K. Christoforou, Alena (Praneet) Ng, Madyson Weippert, Christine Mulligan, Nadia Flexner, Mary R. L’Abbe

**Affiliations:** Department of Nutritional Sciences, Temerty Faculty of Medicine, University of Toronto, Toronto, ON M5S 1A8, Canada

**Keywords:** free sugar, dietary intakes, sugars, calories, reformulation, foods and beverages, Canada

## Abstract

Recommendations suggest limiting the intake of free sugar to under 10% or 5% of calories in order to reduce the risk of negative health outcomes. This study aimed to examine Canadian free sugar intake and model how intakes change following the implementation of a systematic reformulation of foods and beverages to be 20% lower in free sugar. Additionally, this study aimed to examine how calorie intake might be impacted by this reformulation scenario. Canadians’ free sugar and calorie intakes were determined using free sugar and calorie data from the Food Label Information Program (FLIP) 2017, a Canadian branded food composition database, and applied to foods reported as being consumed in Canadian Community Health Survey—Nutrition (CCHS-Nutrition) 2015. A “counterfactual” scenario was modelled to examine changes in intake following the reformulation of foods to be 20% lower in free sugar. The overall mean free sugar intake was 12.1% of calories and was reduced to align with the intake recommendations at 10% of calories in the “counterfactual” scenario (*p* < 0.05). Calorie intake was reduced by 3.2% (60 calories) in the “counterfactual” scenario (*p* < 0.05). Although the overall average intake was aligned with the recommendations, many age/sex groups exceeded the recommended intake, even in the “counterfactual” scenario. The results demonstrate a need to reduce the intake of free sugar in Canada to align with dietary recommendations, potentially through reformulation. The results can be used to inform future program and policy decisions related to achieving the recommended intake levels of free sugar in Canada.

## 1. Introduction

Free sugar (“free sugar”), according to the World Health Organization (WHO), refers to all monosaccharides and disaccharides added to foods by the manufacturer, cook, or consumer, plus sugars naturally present in honey, syrups, fruit juices, and fruit juice concentrates [1]. Evidence demonstrating a link between excessive free sugar intake and adverse health outcomes and unhealthy dietary patterns has led several authoritative health bodies to develop dietary guidelines recommending intake be limited [2,3,4,5]. In 2015, the WHO published a strong guideline recommending that a maximum 10% of calories should be derived from free sugar [1], with a further 5% conditional target, should the strong guidelines be met [1].

Unlike total sugar, which includes both free and naturally-occurring sugars, free sugar levels are not a required declaration on Canadian nutrition labels, nor are they included in the national food composition database, the *Canadian Nutrient File.* Furthermore, free and naturally-occurring sugars are chemically indistinguishable, making the determination of Canadian free sugar intake difficult [6]. There have been some recent efforts made to estimate Canadian free sugar intake. For instance, analyses of the most up to date national nutrition survey data have estimated that free sugar accounts for between 12% and 13.3% of Canadians’ energy intake, and that most Canadians were exceeding the guidelines of a 10% caloric contribution of free sugar [7,8]. These findings align with earlier work, indicating that efforts to improve free sugar intake among Canadians are still needed [9,10,11]. However, these studies relied on data from a generic food database (i.e., the Canadian Nutrient File), which may not accurately represent the specific free sugar levels of products that are sold and consumed in Canada, and to the best of our knowledge, there have been no detailed estimates of the Canadian population-level free sugar intake for all age/sex groups in relation to the WHO intake guidelines.

Reformulating food and beverage products to be lower in free sugar has been proposed as a viable and equitable option to reduce the intake of free sugar, which can benefit the whole population [12]. Given past successes in product reformulation for *trans* fat and sodium in Canada and the UK [13,14], a similar program to lower the free sugar content of foods and beverages is a strong option [12], and has been recently introduced in the UK with the aim of achieving a 20% free sugar reduction [15]. However, free sugar in solid foods contributes to the weight of the product, adding additional complexity to product reformulation [16], which distinguishes free sugar reformulation from other nutrient-focused reformulation situations. In order to maintain the bulk contributed by free sugar, it would need to be replaced either with the addition of another ingredient or through an increase in the proportion of the remaining ingredients. Thus, a reduction in the free sugar content in some foods may not result in an equal reduction in calories [16]. It is therefore essential to determine whether a reformulation of products to be lower in free sugar would result in more or less favorable calorie intake, as calories moderate much of the effects of sugar on chronic disease outcomes [17,18,19].

The objective of this study was to examine the free sugar intake of Canadians overall and by age/sex groups, and to model how intake in free sugar and calories would change if a reformulation strategy to lower free sugar levels by 20% in the Canadian food supply were implemented.

## 2. Materials and Methods

### 2.1. Canadian Community Health Survey—Nutrition (CCHS-Nutrition) 2015

For this study, dietary intakes were calculated using the Canadian Community Health Survey—Nutrition (CCHS-Nutrition) 2015. The CCHS-Nutrition 2015 is a cross-sectional national food and nutrition surveillance survey [20]. Respondents, one per household, included Canadians aged 1 year and older living in private dwellings in the 10 Canadian provinces, and did not include full-time members of the Canadian Forces or those living in some remote areas, institutions (e.g., care facilities and prisons), the Territories, or on reserves and Indigenous settlements [20]. Sampling was carried out to provide a sample representative of the Canadian population in terms of age, sex, geography, and socioeconomic status using a cluster design [20]. CCHS-Nutrition 2015 provides details on food consumption of respondents using 24-h dietary recalls on the total sample from computer-assisted in-person interviews by trained personnel (n = 20,487), and repeat recalls on about 35% of the sample were carried out via telephone [20]. The 24-h dietary recall was based on the Automated Multiple-Pass Method to guide interviewers through the recall so as to maximize opportunities for the respondents to remember and report foods consumed in the previous 24 h [20]. This analysis included the whole CCHS-Nutrition 2015 sample, except for pregnant (n = 119) and breastfeeding (n = 188) women and respondents with null food intake (n = 4), for a total of 20,176 respondents.

### 2.2. Nutrient Composition Used to Determine Free Sugar and Calorie Intake

Free sugar levels are not available in CCHS-Nutrition, thus they were imported for use from the 24-h recall results from a Canadian branded prepackaged food composition database, the Food Label Information Program database (FLIP) 2017 [21]. FLIP has values for free sugar levels calculated primarily using the data available on Canadian food labels, the methods of which have been described previously [21]. Free sugar levels and calorie contents, obtained from the NFt, were merged with the *Food and Ingredient Details* (FID) file used in the CCHS-Nutrition 24-h recall by matching FLIP food products with the equivalent CCHS-Nutrition food profiles, the detailed methods of which have been previously published [22].

Where there was no FLIP match for the CCHS-Nutrition food profiles, usually for whole or fresh foods that would not be captured in a prepackaged food composition database such as FLIP (n = 1224 of 2785 CCHS-Nutrition food profiles), values were imputed from the CCHS-Nutrition food composition database. The calorie levels for the CCHS-Nutrition food profiles were used when there was no FLIP match available and the free sugar levels were calculated primarily using the total sugar levels available in the FID database. For example, for CCHS-Nutrition food profiles that contained 0 g total sugar, the free sugar level was assigned a value of 0 g (n = 454). For CCHS-Nutrition food profiles with total sugars >0 g, the free sugar level was assigned a value reflective of the proportion of total sugar coming from free sugar in products within the same FLIP sugar-focused food category (n = 624). When the CCHS-Nutrition food profile did not have a value for total sugar, the free sugar level assigned was a value reflective of the average free sugar level for products within the same FLIP sugar-focused food category (n = 146). Combined, these values were used to determine the Canadian free sugar and calorie intake.

For the reformulation or “counterfactual” scenario, a 20% reduction in free sugar levels in prepackaged foods was modelled at the level of the food supply [23]. The free sugar levels for all foods were reduced by 20% and the calorie levels were correspondingly adjusted to reflect the changes in calories that accompanied a change in free sugar levels [23]. The rate of calorie level changes was based on the observed difference in calories for products reformulated to be lower in total sugars between 2013 and 2017 by major food categories [23]. Reformulated nutrient compositions for free sugar and calorie levels were also imported for use with the CCHS-Nutrition 24-h recalls.

### 2.3. Statistical Analysis

Free sugar (g per day, % of calories, proportion above 5% of calories from free sugar, and proportion above 10% of calories from free sugar) and calorie intakes (calories per day) were determined using both available days of 24 h-recall from CCHS-Nutrition 2015. The National Cancer Institute (NCI) method was used to obtain the usual intake distributions [24]. Sequence of recalls and whether the recall was carried out for a weekday or weekend day were included as covariates for the NCI method. Univariate analysis was used to determine the absolute free sugar intake and bivariate analysis was used to determine the intake of free sugar as a proportion of calories. Balanced repeated replication (BRR) with 500 replicates, as recommended by Statistics Canada, were used to calculate all of the standard errors using the provided bootstrap weights [25]. All of the analyses were survey-weighted using the master sample survey weights provided by Statistics Canada to allow for representative estimates and were carried out at the Statistics Canada Research Data Centre (Toronto). Estimates of free sugar and calorie intakes were conducted overall and for each DRI age/sex group and were adjusted for age, sex (where not combined), and dietary misreporting status. The misreporting level was defined as a categorical variable with under-reporters identified as those with reported calorie intakes in CCHS-Nutrition 2015 at or less than 70% of their estimated energy requirements (EER) [26], over-reporters identified as those with reported calorie intakes more than 142% of their EER, and plausible reporters identified as those with reported calorie intakes in CCHS-Nutrition 2015 between 70% to 142% of their EER [27,28]. The predicted energy requirements were determined using the Institute of Medicine calculation [26] when measured height and weight or adult self-reported height and weight were available in CCHS-Nutrition 2015. Adult self-reported heights and weights were adjusted using an established correction factor as the existence of bias associated with these measures is known to exist in the Canadian population [29]. For the remaining cases (i.e., where height and weight data were not available for respondents), estimated energy needs for reference heights and weights for each age-sex group were applied using USDA estimates [3]. Physical activity levels for adults were classified as sedentary (i.e., no moderate or vigorous activity or <30 min activity per day), low-active (i.e., moderate or vigorous activity for 30–60 min per day), active (i.e., moderate or vigorous activity for 60–180 min per day), or very active (i.e., moderate or vigorous activity for >180 min per day) [20]. Physical activity levels for children 14 to 19 years old were coded as sedentary, and those 13 years and under were coded as low-active according to previous reports on physical activity in Canadian children [30]. Alternatively, special survey weights that accounted for the non-response of the measured heights and weights were included in CCHS-Nutrition 2015 and could also have been used to evaluate this subsample on its own. To validate our findings, we re-ran the analysis for free sugar intake (g/day) on this subsample only. Overall, the results differed only marginally (1 g mean difference overall). A t-test was used to assess whether the difference in mean free sugar and calorie intakes and intakes in the “counterfactual” scenario were statistically significant overall. Statistical significance levels were set at *p* < 0.05. Analyses were conducted using SAS 9.4 (SAS Institute Inc., Cary, NC, USA).

## 3. Results

### 3.1. Free Sugar Intakes

On average, the reported intakes of free sugar from CCHS-Nutrition 2015 were 56.2 g/per day, ranging from 43.7 g/day among females 71 years or older to 73.5 g/day among children 4 to 8 years and males 9 to 13 years (Table 1). The intakes of free sugar as a proportion of total calories was 12.1% on average, ranging from 10.1% among males 71 years or older to 14.1% among children aged 1 to 3 years (Table 1). Overall, 95.1% had free sugar intakes above 5% of total calories and 60.7% had free sugar intakes above 10% of total calories (Table 1).

### 3.2. Counterfactual Free Sugar Intakes

Canadian free sugar intakes based on the counterfactual scenario, modelling a 20% reduction in free sugar through food and beverage reformulation, were reduced by 11.3 g/day on average to 44.9 g/day (*p* < 0.05) (Table 2). Free sugar intakes were significantly lower in the “counterfactual” scenario overall and for each age/sex group (Table 2). The percentage of total calories from free sugar were reduced by 17.4% to an average of 10% overall (*p* < 0.05) (Table 2). The proportion of the Canadian population with free sugar intakes above 5% of the total calories went down to 90.4% and intakes above 10% of the total calories went down to 43.6% (*p* < 0.05) (Table 2).

### 3.3. Counterfactual Calorie Intakes

Overall, intakes were 60 calories (or 3.2%) lower in the “counterfactual” scenario among the whole population (*p* < 0.05) (Table 3). Calorie intakes in the “counterfactual” scenarios were similarly approximately 3% lower for each age/sex group (Table 3).

## 4. Discussion

This study examined the free sugar intake of Canadians overall and by age/sex groups, and was the first to model how the intake in free sugar and calories would change if a reformulation strategy to lower free sugar levels by 20% in the Canadian food supply was implemented. The present study used a novel approach to examine the free sugar intake, using branded label data from a national database of pre-packaged foods and beverages sold in Canada (FLIP) [31], along with data from the generic CNF database for unprocessed products. Using such an approach allowed us to model the impact of replacing top sources of free sugar by their lower sugar counterparts (if available) using actual foods from the Canadian marketplace, enabling an analysis of the impact of sugar reformulation not only on sugar intake, but also its impact on energy intake. This study demonstrates a high intake of free sugar among Canadians, with all age/sex groups exceeding the maximum recommendation of 10% of calories from free sugar. The findings predict that the resultant mean intakes would be in line with the WHO 10% recommendation following a strategic and systematic effort to reformulate the food supply to be 20% lower in free sugar levels; however, several age/sex groups would still have an excessive intake, and the intake of calories would be reduced marginally or not at all.

The mean Canadian intake of free sugar exceeded the strong guidelines from the WHO overall and for every age/sex group [1], which was aligned with research conducted in Canadian children [32]. Not only were the mean intakes above the WHO strong guideline, but about 60% of Canadians in this study exceeded this guideline and almost all exceeded the conditional guideline of 5% of calories from free sugar [1]. The free sugar intake tended to be the highest among the younger age groups compared with the older ones, which is consistent with research on total sugar intake [33], perhaps indicating that children and adolescents are particularly at risk of an overconsumption of free sugar. Evidence has demonstrated a link between the excessive consumption of free sugar and the risk of dental caries, obesity, type 2 diabetes, and cardiovascular disease [18,19,34,35,36]. It is apparent from the results of this study that for intakes to align with the WHO guidelines, a reduction in Canadian free sugar intake is needed.

Modelling the food supply to be 20% lower in free sugar resulted in an equivalent reduction in free sugar intake, bringing the mean overall intakes in line with the WHO strong recommendation. The % difference for free sugar intake, measured as grams per day and as % of calories, varied slightly from one another, with an average 20.1% reduction in free sugar (g/day) but only an average of 17.4% reduction in free sugar as a % of calories. This may be a result of the latter measure being reflective not only of free sugar intake, but also of caloric intake, which fluctuates according to the intake of other nutrients. Despite the overall reduction in free sugar intake, the intakes for the “counterfactual” scenario still tended to be higher among the children and adolescents compared with other age groups. This finding may indicate that additional efforts to target specific sub-populations may be needed; for instance, recent efforts attempting to restrict the marketing of foods high in sugars, sodium, and saturated fats to children [37] or the implementation of sugar sweetened beverage taxes, given the high consumption of such products by adolescents and young people [38,39].

Calorie intake mediates some of the adverse health outcomes associated with free sugar intake, particularly those related to body weight [17,18,19]. Thus, it was imperative to examine not only the change in free sugar intake that would accompany a reformulation of the food supply, but also the change in calorie intake. Overall, the intake in the “counterfactual” scenario was 60 calories or 3.2% lower than the “actual” scenario, much lower than the reduction seen in the free sugar intake. This small difference, however, was not only statistically significant, but could also be nutritionally significant. In fact, a reduction in calorie intake of only 100 calories per day was recommended to achieve moderate weight loss by the UK’s *Calorie Reduction Expert Group* [40]. The percentage reduction in calorie intake tended to be similar across different age/sex groups, all ranging between 2.9% and 3.5%. At an absolute level, the difference in calorie intake (60 calories) varied by more than the equivalent difference in calorie contribution from free sugar (11.3 g free sugar is equivalent to 45.2 calories). This discrepancy may be a result of the direction and magnitude in which calories change, with a 20% reduction in free sugar levels of foods and beverages. Further investigation of the main food and beverage contributors to free sugar intake could further elucidate the findings presented in this study to explore differential rates of consumption and explain which foods and beverages may have contributed to the calorie reduction.

This study has several strengths and limitations. FLIP 2017 is a representative database of prepackaged foods and beverages available for sale in Canada, and previous iterations of FLIP have included 93.8% of foods and beverages sold throughout all banner stores of the largest grocery retailer in the country (data not published). In order to keep subjectivity to a minimum when matching FLIP products with food profiles used in the CCHS-Nutrition food composition database, a step-by-step decision-making process was outlined, a second researcher validated a subsample of matches, and discrepancies were identified and resolved through consensus between researchers [22]. In light of the absence of free sugar levels on Canadian nutrition labels and in the food composition data used in CCHS-Nutrition, the free sugar of FLIP products was estimated using a decision-making algorithm [21]. This algorithm allowed for the first calculations of free sugar levels in Canadian foods and this systematic methodology has been shown to have high levels of inter-researcher repeatability [41]. Using the FLIP database as a source of nutrient information enabled robust estimates of Canadian population-level free sugar intake based on comprehensive and contemporary food composition data using the individual food profiles reported as being consumed in CCHS-Nutrition, while previous estimates of added sugar intake were limited to assumptions based on the reported consumption of food groups or generic food composites [11]. To the best of our knowledge, this is the first study to model the impact of a systematic free sugar reduction scenario on Canadian intake and it uses an existing reformulation strategy, the UK free sugar reduction target of 20%, as its basis [15]. Additionally, this modelling was based on actual observed changes in nutrient levels, including calories, seen in foods reformulated to be lower in sugar between 2013 and 2017 [42]. In modelling the “counterfactual” scenario, the present study assumed that food choice would remain the same regardless of a reduction in free sugar level for foods and beverages. Food choice is complex and many potentially competing factors influence selection and intake [43]. Taste is the main factor contributing to Canadian food choice [44], thus, it is possible that intake would be altered if product formulation and subsequently taste changed. Should such a free sugar reduction strategy be implemented in Canada, investigations regarding how purchases and intakes may be further affected would be imperative. Lastly, CCHS-Nutrition is nationally representative and survey sampling and 24-recall collection are rigorous. Dietary intake can often be reported with bias; however, attempts to limit this were built into the CCHS-Nutrition study design using the Automated Multiple Pass Method, which minimizes misreporting bias [20]. To further account for this bias, a variable for potential misreporting was used a covariate in the present study. This study also used the NCI method to estimate the usual intake from CCHS-Nutrition [24]. The NCI method can adjust for between-person variation (e.g., age, sex, and dietary misreporting) and within-person variation in the case of bivariate analyses, and mitigates some of the limitations of using cross-sectional data [24].

## 5. Conclusions

The findings from this study provide comprehensive estimates of Canadian free sugar intakes using a food composition database of brand name foods on the Canadian market and support the need to reduce intakes to better align with recommendations from the WHO. Furthermore, this study demonstrates that the reformulation of foods and beverages to be lower in free sugar could result in equivalent reductions in free sugar intakes as well as lower, and possibly nutritionally significant, intakes of calories. Results can be used to inform future program and policy decisions related to lowering the intakes of free sugar in Canada.

## Figures and Tables

**Table 1 foods-12-01771-t001:** Canadian free sugar intakes presented as grams/day, % of calorie intake (TE), and the proportion of with intakes above 5% and 10% of calories from free sugar, overall and by age/sex group (n = 20,176).

Age/Sex Group	n	Mean (SE) (g/day)	P25 (g/day)	P50 (g/day)	P75 (g/day)	Mean (SE) %TE	P25 (%TE)	P50 (%TE)	P75 (%TE)	% >5% TE (SE)	% >10% TE (SE)
1 to 3 years	1324	67.3 (1.5)	38.9	59.1	86.7	14.1 (0.2)	10	13.2	17.3	98.0	74.8
4 to 8 years	1233	73.5 (1.4)	46.3	67.2	93.4	13.9 (0.2)	10	13.1	16.9	98.4 (0.3)	74.9 (1.5)
9 to 13 years (m)	1047	73.5 (1.5)	46	66.8	93.9	13.3 (0.2)	9.6	12.6	16.3	98.0 (0.4)	71.6 (1.4)
9 to 13 years (f)	969	62.8 (1.5)	38.3	56.6	80.5	14.0 (0.2)	9.9	13.2	17.1	98.0 (0.4)	74.4 (1.5)
14 to 18 years (m)	960	71.2 (1.7)	43.3	64.4	91.4	13.0 (0.2)	9.4	12.3	15.9	97.6 (0.5)	69.9 (1.4)
14 to 18 years (f)	1031	56.7 (1.4)	33.1	50.4	73.3	13.6 (0.2)	9.5	12.8	16.8	97.3 (0.5)	71.4 (1.6)
19 to 30 years (m)	882	63.9 (1.7)	38.7	57.4	82.1	12.6 (0.2)	8.9	11.8	15.5	96.9 (0.5)	65.9 (1.4)
19 to 30 years (f)	897	49.9 (1.3)	28.9	44.5	64.9	13.2 (0.2)	9.1	12.3	16.3	96.4 (0.6)	67.7 (1.5)
31 to 50 years (m)	2077	60.2 (1.0)	36.7	54.2	77.2	11.8 (0.1)	8.3	11.1	14.6	95.4 (0.6)	59.9 (1.3)
31 to 50 years (f)	2288	48.2 (0.9)	27.7	42.6	62.6	12.2 (0.2)	8.3	11.4	15.2	95.1 (0.7)	61.3 (1.4)
51 to 70 years (m)	2246	56.4 (1.1)	33.1	50.1	73	11.0 (0.1)	7.6	10.3	13.5	93.6 (0.8)	52.8 (1.3)
51 to 70 years (f)	2420	46.8 (0.9)	26.9	41.3	60.6	11.3 (0.2)	7.7	10.6	14.1	93.2 (0.9)	55.1 (1.4)
71 years or older (m)	1246	53.5 (1.2)	31.9	48	69	10.1 (0.2)	7	9.5	12.6	91.4 (0.9)	45.2 (1.5)
71 years or older (f)	1556	43.7 (1.0)	24.7	38.5	57	10.4 (0.2)	7	9.7	13	90.5 (1.1)	47.3 (1.6)
**All**	**20,176**	**56.2 (0.6)**	**32.4**	**49.7**	**72.8**	**12.1 (0.1)**	**8.3**	**11.3**	**15**	**95.1 (0.6)**	**60.7 (1.1)**

All analyses were adjusted for age, sex (where not combined), and misreporting using the National Cancer Institute (NCI) method was used to obtain usual intake distributions [24]. Abbreviations: f = females; g = grams; m = males; SE = standard error; TE = total energy (calories).

**Table 2 foods-12-01771-t002:** Counterfactual ^a^ Canadian free sugar intakes presented as grams/day, % of calorie intake (TE) ^b^, and the proportion of with intakes below 5% and 10% of calories from free sugar, overall and by age/sex group (n = 20,176).

		Counterfactual Free sugar Intake				Absolute Difference ^c,d^				% Difference ^e^			
Age/Sex Group	n	g/day	%TE	Above 5% TE (%)	Above 10% TE (%)	g/day	%TE	Above 5% TE (%)	Above 10% TE (%)	g/day	%TE	Above 5% TE (%)	Above 10% TE (%)
1 to 3 years	1324	53.8 (1.2)	11.7 (0.2)	95.8 (0.6)	59.0 (1.5)	13.5	2.4	2.2	15.8	20.1%	17.0%	2.2%	21.1%
4 to 8 years	1233	58.8 (1.1)	11.5 (0.1)	96.2 (0.6)	58.5 (1.5)	14.7	2.4	2.2	16.4	20.0%	17.3%	2.2%	21.9%
9 to 13 years (m)	1047	58.8 (1.2)	11 (0.2)	95.5 (0.7)	54.7 (1.5)	14.7	2.3	2.5	16.9	20.0%	17.3%	2.6%	23.6%
9 to 13 years (f)	969	50.2 (1.2)	11.6 (0.2)	95.6 (0.7)	58.9 (1.6)	12.6	2.4	2.4	15.5	20.1%	17.1%	2.4%	20.8%
14 to 18 years (m)	960	56.9 (1.3)	10.8 (0.1)	94.7 (0.7)	52.1 (1.4)	14.3	2.2	2.9	17.8	20.1%	16.9%	3.0%	25.5%
14 to 18 years (f)	1031	45.4 (1.1)	11.3 (0.2)	94.4 (0.8)	55.6 (1.5)	11.3	2.3	2.9	15.8	19.9%	16.9%	3.0%	22.1%
19 to 30 years (m)	882	51.1 (1.4)	10.5 (0.1)	93.4 (0.8)	48.1 (1.3)	12.8	2.1	3.5	17.8	20.0%	16.7%	3.6%	27.0%
19 to 30 years (f)	897	39.9 (1.0)	10.9 (0.1)	93.2 (0.8)	51.6 (1.3)	10.0	2.3	3.2	16.1	20.0%	17.4%	3.3%	23.8%
31 to 50 years (m)	2077	48.1 (0.8)	9.8 (0.1)	90.6 (0.9)	42.1 (1.2)	12.1	2.0	4.8	17.8	20.1%	16.9%	5.0%	29.7%
31 to 50 years (f)	2288	38.5 (0.7)	10.1 (0.1)	90.6 (1.0)	44.8 (1.2)	9.7	2.1	4.5	16.5	20.1%	17.2%	4.7%	26.9%
51 to 70 years (m)	2246	45.1 (0.9)	9.0 (0.1)	87.7 (1.1)	34.7 (1.2)	11.3	2.0	5.9	18.1	20.0%	18.2%	6.3%	34.3%
51 to 70 years (f)	2420	37.4 (0.7)	9.4 (0.1)	87.5 (1.2)	37.8 (1.2)	9.4	1.9	5.7	17.3	20.1%	16.8%	6.1%	31.4%
71 years or older (m)	1246	42.8 (0.9)	8.4 (0.1)	83.8 (1.3)	27.7 (1.4)	10.7	1.7	7.6	17.5	20.0%	16.8%	8.3%	38.7%
71 years or older (f)	1556	34.9 (0.8)	8.6 (0.1)	83.2 (1.4)	30.6 (1.3)	8.8	1.8	7.3	16.7	20.1%	17.3%	8.1%	35.3%
**All**	**20,176**	**44.9 (0.5)**	**10.0 (0.1)**	**90.4 (0.9)**	**43.6 (0.9)**	**11.3**	**2.1**	**4.7**	**17.1**	**20.1%**	**17.4%**	**4.9%**	**28.2%**

^a^ Counterfactual scenario models a 20% free sugar reduction through food and beverage reformulation. ^b^ Means (SE) adjusted for age, sex, and misreporting using the National Cancer Institute (NCI) method was used to obtain usual intake distributions [24]. ^c^ Absolute difference calculated by subtracting the “counterfactual” intakes from the original intakes (Table 1). ^d^ Difference between the mean original intakes and “counterfactual” intakes was statistically significant overall and for each individual age/sex group (*p* < 0.05). ^e^ % Difference calculated by dividing the absolute difference by the original intakes (Table 1). Abbreviations: f = females; g = grams; m = males; SE = standard error; TE = total energy (calories).

**Table 3 foods-12-01771-t003:** Canadian calorie intakes (calories/day) ^a^ for the “actual” and “counterfactual” ^b^ scenarios, overall and by age/sex group (n = 20,176).

Age/Sex Group	n	Actual Scenario	Counterfactual Scenario	Absolute Difference ^c^	% Difference ^d^
1 to 3 years	1324	1919 (39)	1852 (37)	67	3.5%
4 to 8 years	1233	2121 (30)	2048 (29)	73	3.4%
9 to 13 years (m)	1047	2209 (31)	2135 (30)	74	3.3%
9 to 13 years (f)	969	1806 (30)	1743 (29)	63	3.5%
14 to 18 years (m)	960	2181 (39)	2108 (38)	73	3.3%
14 to 18 years (f)	1031	1664 (30)	1606 (29)	58	3.5%
19 to 30 years (m)	882	2023 (44)	1956 (43)	67	3.3%
19 to 30 years (f)	897	1515 (35)	1462 (34)	53	3.5%
31 to 50 years (m)	2077	2037 (26)	1972 (25)	65	3.2%
31 to 50 years (f)	2288	1567 (22)	1515 (21)	52	3.3%
51 to 70 years (m)	2246	2054 (29)	1991 (29)	63	3.1%
51 to 70 years (f)	2420	1642 (19)	1590 (19)	52	3.2%
71 years or older (m)	1246	2091 (26)	2030 (26)	61	2.9%
71 years or older (f)	1556	1658 (24)	1608 (24)	50	3.0%
**All ***	**20,176**	**1858 (12)**	**1798 (11)**	**60**	**3.2%**

* Indicates a statistically significant difference between the overall mean intake and “counterfactual” intake (*p* < 0.05). ^a^ Means (standard error) adjusted for age, sex, and misreporting using the National Cancer Institute (NCI) method was used to obtain usual intake distributions [24]. ^b^ Counterfactual scenario models a 20% free sugar reduction through food and beverage reformulation. ^c^ Absolute difference calculated by subtracting the “counterfactual” intakes from the original intakes. ^d^ % Difference calculated by diving the absolute difference by the original intakes. Abbreviations: f = females; m = males.

## Data Availability

Data is available upon reasonable request from the authors.

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
