# Peer review of "Canadian Free Sugar Intake and Modelling of a Reformulation Scenario"

_foods, 2023, doi:10.3390/foods12091771_

Round 1

Reviewer 1 Report

Thank you for the opportunity to review this well-written manuscript which focuses on free sugar intake and the predicted effect of reformulation of foods per a modeling technique.  The details regarding the development of the "reformulated" foods and their estimated free sugar intake is well-described.  They conclude that while reformulation can possibly reduce free sugar intake by a clinically beneficial amount further research would be needed to incorporate the other aspects of reformulation such as taste.

Minor concerns:

1.  A reference should be provided for the balanced repeated replication noted on line 134.  

2.  Line 139 states that estimates were adjusted for age, sex and misreporting, but line 163 states that t-tests were used for comparison.  Additional clarity on how the adjustments were used would clear up this apparent inconsistency.

3.  Tables 1 & 2 are extremely busy and difficult to read.   In Table 1, it's not clear what the "25th", "50th" and "75th" columns for both grams/day and %TE are referring to.  It would appear quartiles, but then what is the SE?  Since these values are not referenced to in the Results or Conclusions, I would suggest moving them to an Appendix along with a footnote stating what they represent.

4.  With the magnitude of t-tests performed, an adjustment to the p-values is warranted, especially since specific p-values are not presented and are denoted as "p<0.05".

Reviewer 2 Report

The main aim of paper: “Canadian Free Sugar Intakes and Modelling of a Reformulation Scenario” was to examine the free sugar intakes of Canadians overall and by age/sex groups and to model how intakes in free sugars and calories would change if a reformulation strategy to lower free sugar levels by 20% in the Canadian food supply were implemented.

From my point of view, the topic of the study is important and really interesting.

In general, the procedure, the study design and statistical analyses are well organized.

Moreover, the topic is quite original and the obtained results in general confirmed that the reformulation is needed.

However, I would like to ask you about the consumer behaviour in this area. Are consumers willing to accept these potential changes? What is your opinion in this field? This reformulation is needed because of the nutritional and health benefits. This is obvious for the nutrition specialists and for the food technologists but I am afraid that consumers will react differently (maybe they will not accept those changes) because of their preferences.

Round 2

Reviewer 2 Report

Dear Authors,

Thank you for your response.

Kind regards

Author Response

We thank the reviewer for their efforts.